# Distance interventions for enhancing preparedness in informal caregivers of older adults: A systematic review protocol

**Fernanda L. F. Dal Pizzol**[1]*, **Kathleen F. Hunter**[1‡], **Jennifer Baumbusch**[2], **Hannah M. O'Rourke**[1‡]

1 Faculty of Nursing, College of Health Sciences, University of Alberta, Edmonton Clinic Health Academy, Edmonton, AB, Canada, 2 School of Nursing, University of British Columbia, Vancouver, BC, Canada

‡ These authors contributed equally to this work. KFH and HMO are Joint Senior Authors
* fenglerd@ualberta.ca

## Abstract

**Data Availability Statement:** Deidentified research data will be made publicly available when the study is completed and published.

### Introduction

Informal caregivers provide care to older adults but report lack of preparedness to enact the role. Intervention programs delivered by distance offer one alternative to support preparedness. Three review studies conducted to date have highlighted the benefits of distance interventions for enhancing preparedness among informal caregivers of older adults. However, these reviews have been limited in presenting and discussing how intervention components influenced outcomes. Additionally, they have not compared different distance delivery approaches for informal caregivers of older adults or assessed their varying impact on preparedness outcomes. These limitations make the effectiveness of diverse distance approaches unclear.

### Aim

To evaluate the effects of distance interventions aimed at enhancing preparedness among informal caregivers of older adults.

### Methods

This protocol follows the Preferred Reporting Items for Systematic Reviews and Meta-Analyses Protocols (PRISMA-P) guidelines and is guided by the Cochrane Handbook for Systematic Reviews of Interventions. It has been registered in PROSPERO (CRD42023400668). Databases used in the search will include CINAHL, Cochrane Library, EMBASE, MEDLINE, PsycINFO, Scopus, and ProQuest Theses and Dissertations Global. The search will not be restricted by publication year to include all relevant studies. Studies published in English and Portuguese will be included. Study quality will be assessed using Downs and Black's checklist. If metanalysis is possible, it will be performed using the ReviewManager (RevMan) software.

**Funding:** Fernanda Dal Pizzol, PhD Candidate, is a recipient of the 2024 Alberta SPOR Graduate Studentship in Patient-Oriented Research. The Alberta SPOR Graduate Studentships in Patient-Oriented Research are jointly funded by Alberta Innovates and the Canadian Institutes of Health Research.

## Conclusions

The study will be the first of its type to systematically review and synthesize components and approaches of distance interventions aimed at supporting preparedness of informal caregivers of older adults.

## 1. Introduction

Informal caregiving refers to the care provided by family members, friends, and neighbors to a care-dependent person, meaning someone who depends on others for the fulfillment of their daily activities [1,2]. It is estimated that approximately 30% of care-dependent individuals are over 60 years of age, with informal caregivers commonly serving as the primary source of support for this population in most countries [3]. In addition to performing domestic duties, informal caring entails a variety of responsibilities and areas of concentration, including practical, medical, emotional, and social support [4]. However, informal caregivers often feel unprepared for providing care [5,6]. Scholars have identified that lack of preparation can negatively impact the lives of caregivers, care recipients, and the health care system [7–10].

Caregivers' lack of access to support services may present challenges that can impact not only the care recipient's actual health treatment but, more broadly, their ability to take care of their own health and well-being [11,12]. The reasons why people do not have access to health services are related, for example, to geographic location, accessibility to transport, and presence of multiple health conditions [13,14]. In this context, support provided by distance (e.g., online) emerges as an important option in providing caregiver support [15], with studies indicating that distance interventions are effective in supporting preparedness of informal caregivers [16–18]. However, it is important to assess which components (i.e., goal, content/activities) have been used in distance interventions aimed at supporting preparedness in order to understand what makes the interventions effective (or not).

## 2. Background

In a rapidly aging global population where the number of older adults is significantly increasing [19], the role of family and friends as caregivers has become increasingly vital [20]. However, it is often the case that these caregivers do not receive adequate support [2,11]. This lack of support is particularly pronounced in many regions where resources for caregiver assistance are still in the early stages of development or implementation, such as rural and remote areas and less developed countries [21–23]. Given this context, it becomes imperative to focus on preparing these caregivers for their role [24]. By investing in caregiver preparation programs, we not only empower caregivers to provide better care but also contribute to the overall well-being of older adults and their families [24–26]. Ultimately, prioritizing caregiver preparation is not only a matter of meeting immediate caregiving and older adult needs but also an investment in the long-term sustainability of care for older adults [24–26].

Preparedness for caregiving has been defined in a variety of ways over time [27], and preparedness of caregivers of older adults is currently defined as "caregiver's self-confidence about their current competence related to the knowledge, skills and abilities to perform daily tasks, and to handle emotions over time" [19, p. 2318]. Several studies have directly evaluated preparedness, for example, using the Preparedness for Caregiving Scale [28], which is a self-report instrument that asks caregivers how well prepared they believe they are for multiple

domains of caregiving, including physical care, providing emotional support, setting up in-home support services, and dealing with the stress of caregiving [7,29–31]. Additionally, other studies have used multiple surrogate terms or related concepts to assess caregiver preparedness [27], including: readiness [32,33], confidence [34–36], competence [37], capacity [38,39], and self-efficacy [28,32,40]. These surrogate terms and related concepts share common antecedents and attributes with preparedness, such as knowledge, skills, and abilities to perform daily tasks, as well as self-confidence. However, they also exhibit distinct features, particularly a notable focus on the quantitative aspect of preparedness and a lack of comprehensive explanation of its development [27]. For example, readiness is defined as "being available and ready to act at any time when the older person returns from hospital to home (. . .) putting one's own needs aside to stand up for the older person whenever required' [41 p. 7]". It shares a common attribute with preparedness (performing daily tasks) and an antecedent (having the time to adjust routines and being available to provide care at any time) [27]. However, it differs from preparedness because its definition does not fully clarify the process of how preparedness occurs [27].

Distance interventions are one alternative to support caregiver's preparedness and may promote accessible support [15]. Numerous in person interventions designed to support preparedness have been identified as effective for use with family caregiving in oncology and dementia [42–46]. However, delivering such interventions in person presents challenges, particularly concerning accessibility due to distance or transportation issues. Consequently, distance support has emerged as a feasible alternative and a valuable complement to traditional in person supports [15]. Distance support has become increasingly attractive in the last decade due to the widespread use of personal computers and the adoption of online learning in elementary to post-secondary school systems [47], particularly following the COVID-19 pandemic [48].

Technology-based interventions have been deemed acceptable by informal caregivers and offer numerous advantages, benefiting both caregivers and health professionals [15,47–49]. Technology-based interventions utilize information and communication technologies to help overcome distance barriers and facilitate scheduling logistics, thereby expanding the scope for delivery of high-quality healthcare [50]. Technological platforms can facilitate informal caregivers in expressing their concerns [51] and health professionals in supporting caregivers by providing services such as discharge follow-up [52]. The potential advantages of supporting caregivers by distance using technology include reduction of time and costs (e.g., travel, accommodation, hospitality) for health professionals [49], reduction of time for caregivers and time away from the care recipient [53], and convenience for both participants and health professionals, as support can occur in more accessible locations [13,14]. According to the 2016 American Association of Retired Persons nationally representative survey of caregivers [47], 71% of caregivers are interested in the use of technology to support their caregiving tasks and 59% report they are likely to use a currently available technology.

## 2.1 Why is this review needed in light of existing reviews?

To date, three literature review studies have examined distance interventions targeting the preparedness of informal caregivers of older adults. These reviews suggest that distance approaches can effectively support caregiver preparedness [16–18]. However, these reviews primarily focused on studies involving both formal and informal caregivers, overlooked telephone sessions as a technological tool, and did not exclusively consider distance interventions approaches in all studies. As a result, our understanding of the effects of distance interventions on informal caregiver preparedness remains limited. For instance, telephone sessions, a

recognized distance approach, have demonstrated efficacy in enhancing caregiver preparedness [54], and formal caregivers often possess prior preparedness acquired through formal education.

Furthermore, the previous reviews offered minimal discussion on how intervention components, such as dosage and specific activities performed during the intervention, influenced outcomes. They also failed to compare various distance approaches available for informal caregivers or assess their distinct impacts on outcomes related to caregiver preparedness. These gaps leave the effectiveness of distance approaches unclear. Our systematic review aims to fill these gaps by examining the effectiveness of distance interventions by intervention component.

Future reviews should explore how specific components of distance interventions, such as educational materials and emotional support, influence outcomes. Additionally, they should compare different distance approaches targeting informal caregivers of older people. To provide a comprehensive and unbiased synthesis of the evidence [55], this proposed systematic review will evaluate the effects of diverse distance interventions aimed at enhancing the preparedness of informal caregivers of older adults.

## 3. Aim

The purpose of this systematic review will be to evaluate the effects of distance interventions aimed at enhancing preparedness among informal caregivers of older adults.

### 3.1 Systematic review question

What are the effects of diverse distance interventions that aim to enhance preparedness of informal caregivers of older persons?

## 4. Methods

This protocol is reported according to the Preferred Reporting Items for Systematic Review and Meta-Analysis Protocols (PRISMA-P) guidelines [56] (see S2 Appendix).

### 4.1 Inclusion and exclusion criteria

The inclusion criteria are: a) studies participants are informal caregivers (18 years of age or older, of all genders) of older adults (60 years of age or older, of all genders); b) studies must report quantitative findings related to the effects of an entirely distance intervention (e.g., by phone, videoconference); c) studies will be included if they are experimental or quasi-experimental; d) quasi-experimental studies must have a comparator (any control group such as placebo, usual care, active control, single pre-post); d) preparedness must be an intervention outcome, either as a primary or secondary outcome; and e) the intervention study must have measured caregiver preparedness as an outcome, using a scale (e.g., preparedness for caregiving scale, capacity scale, confidence scale) or with a single item.

No limits will be defined for the place of residence, recruitment site (e.g., community-based, during hospitalization), and chronicity (e.g., dementia, stroke) of the care recipient, reflecting the diversity of informal caregiving. In addition, the years of publication will not be limited, to include all studies that meet the inclusion criteria. The full text for included studies must be in English or Portuguese, both languages the author and the second reviewer are able to read and comprehend. S1 Appendix details the "Inclusion and Exclusion Form" for this study.

## 4.2 Search protocol

A health sciences librarian was consulted to identify relevant databases. Seven electronic databases will be included: CINAHL, Cochrane Library, EMBASE, MEDLINE, PsycINFO, Scopus, and ProQuest Theses and Dissertations Global. These databases are good resources for identifying relevant publications as one is more focused on nursing (CINAHL) and the others cover health research more broadly. The decisions regarding the keywords and Medical Subject Headings (MeSH) to be used in the literature search were identified through the review of studies on informal caregivers' preparedness, in consultation with a health sciences librarian, and by considering their reflection and alignment with our selection criteria. The keywords and MeSH terms that will be used in this study will include the concepts of measurement (e.g., measure, scale), preparedness (e.g., preparedness, readiness), informal caregivers (e.g., informal, caretaker), and distance learning (e.g., remote, virtual). The concepts of preparedness included in our search strategy encompass surrogate terms (e.g., readiness), related concepts (e.g., competence, capacity), or elements of the preparedness concept (e.g., mutuality). We adopt this approach to ensure that important studies examining preparedness, albeit not explicitly labeled as such, are not overlooked. The specific data-based search strategies are available in S3 Appendix. Our search was conducted in December 2023, and we will update it six to 12 months before publication, in accordance with Cochrane recommendations [57].

Regarding grey literature, we plan to include empirical studies that have essentially measured preparedness. This will involve including theses and dissertations, which will be sourced from ProQuest Theses and Dissertations Global. Additionally, we will conduct a backwards and forwards citation searching [58] by utilizing *citationchaser* tool [59] to identify additional eligible primary studies. Forward citation searching will locate studies that cite the included studies, while backward citation searching will identify studies referenced by the included studies [58].

## 4.3 Reference and data management

All titles and abstracts retrieved from electronic databases and through citation searching will be directly exported to an online platform that supports evidence synthesis projects (i.e., Covidence) using files in RIS format. Covidence will be utilized for deduplication, organizing the screening process, and tracking reasons for excluding full text studies. Data will be extracted into a excel and/or word files. Mendeley will be used as a citation manager as it has free access. All the studies that were screened in full text will be uploaded to a folder in Mendeley to ensure accurate lists of included and excluded studies.

## 4.4 Screening trials

The first 25 studies will be screened (titles and abstracts) through a trial, where three reviewers will independently review the records. After screening, reviewers will convene to verify their agreement regarding inclusions/exclusions, to address any potential questions, and to discuss if there are any further clarifications required to the screening form. In addition, the three reviewers will independently review the first five studies (full text). Subsequently, the review team will meet to discuss any disagreements and provide clarification if needed. A fourth reviewer, with experience in conducting systematic reviews will be consulted in the event of any disagreement among the reviewers.

## 4.5 Screening

Two review authors will independently screen all the titles and abstracts and full text studies for inclusion. Reasons for excluding studies will be documented during the full text review

using Covidence and recorded in a Preferred Reporting Items for Systematic reviews and Meta-Analyses (PRISMA) flow diagram [60]. Any disagreements will be resolved through discussion or, if required, by consulting other review team members.

## 4.6 Data extraction

Data will be tabulated in Microsoft Excel and/or Word spreadsheets. A pilot data extraction will be conducted by three reviewers to discuss the operationalization of the data extraction tables and instructions. Following this, one review author will extract information from the included studies, and a second reviewer will verify the extraction. The extracted data will include:

- author(s) and year of publication

- concept/framework: concept of preparedness used and framework/theoretical foundation that guided the study

- study setting: country and setting (hospital, primary health care service)

- characteristics of caregivers (age, biological sex, relationship with the older person, educational level, ethnicity, employment status, income)

- information about caregiving (length as a caregiver, support from others, out of pocket expenses, hours per week providing care)

- characteristics of older persons (age, comorbidity, level of dependency)

- study design and sample (design, eligibility criteria, number of participants in each group, number of participants excluded)

- intervention (goal, components and activities details, mode of delivery, dose, definition of comparison, timing, randomization, resources used)

- outcome measures (tool or instruments, association with other variables, timing points, outcome assessor)

- results for the effects upon preparedness (significance of effect, baseline and last follow-up mean values, mean difference, standard deviation, risk ratio and odds ratio)

Where multiple reports of the same trial are retrieved, data will be extracted from the multiple reports and grouped together in a single entry in the data extraction table to ensure complete extraction and avoid counting the same study multiple times. Any disagreement regarding data extraction will be resolved through discussion or by involving other review team members.

## 4.7 Quality appraisal

We will use Downs and Black's checklist [61] to appraise study quality. This checklist has been widely used in systematic review studies [62–66]. This tool was built using a specified empirical approach and is a valid and reliable checklist [62–67]. Due to the anticipated heterogeneity of the primary studies to be included in this review, this checklist is appropriate as it can assess both randomised and non-randomised studies.

Twenty-seven items (covering areas such as reporting, external and internal validity, and power) are rated either as yes (= 1) or no/unable to determine (= 0), and one item is rated on a 3-point scale (yes = 2, partial = 1, and no = 0). Scores range from 0 to 28, with higher scores indicating studies of higher methodological quality. Suggested cut-off points are: excellent 26–28; good 20–25; fair 15–19; and poor ≤14 [68].

A pilot use of the quality appraisal tool will be conducted with at least one study. Three review authors will independently appraise the study using the tool. Subsequently, they will convene to discuss their ratings and ensure consistent application of the criteria [69]. If disagreements arise, they will seek consultation from an experienced reviewer.

Two review authors will independently assess quality for each included study. Any disagreements will be resolved through discussion or by involving the review team. To support the judgments made, the reviewers will include information in appraisal tables, producing a summary table demonstrating how each study rated on each item [69]. Direct quotes from study reports detailing the methods employed, as well as an explanation of why a particular method is flawed, will be included in this table as needed.

The conclusions drawn in a systematic review depend on the results of the included studies. Thus, if these results are biased in the primary studies, then the review will produce a misleading conclusion [69]. This review will consider the risk of bias in the results of the included studies when interpreting the results of this review. In addition, the quality of the studies will be assessed to guide clinicians, researchers, and other collaborators for decision-making based on the quality assessment and recommendation of this review.

## 4.8 Data analysis

We will use Microsoft Excel to generate frequencies and descriptive statistics for the total number of included studies. Types of study designs, countries, and settings where studies were conducted, and characteristics of the target population (e.g., biological sex, relationship with the older adult) will be presented. The analysis of intervention effects will depend on whether a meta-analysis is possible. A meta-analysis is the statistical combination of results from two or more separate studies and has numerous advantages [70]. However, there are circumstances where undertaking a meta-analysis may not be possible, such as the clinical and methodological diversity of the included studies, which is likely in this review. In such cases, alternative statistical synthesis methods may need to be considered [71].

**4.8.1 Identifying and measuring heterogeneity.**   One reviewer will assess clinical and methodological diversity (i.e., whether populations, interventions, outcome measures, and study designs are sufficiently similar conceptually) [70] and statistical heterogeneity, in consultation with the review team. Statistical heterogeneity will be assessed through the Cochran Q test and $I^2$ statistic, which describes the percentage of the variability in effect estimates [70,72,73]. A p value $< 0.1$ or an $I^2$ value $> 50$ percent represents substantial or considerable statistical heterogeneity [70,72,73]. If there is limited statistical heterogeneity (p value $>0.1$ or an $I^2$ value $< 49$ percent), and the populations, interventions, outcome measures, and study designs are judged by the team to be sufficiently consistent and conceptually homogenous, then a meta-analysis will be conducted.

**4.8.2 When a meta-analysis of effect estimates is possible.**   ReviewManager (RevMan) software will be used to perform meta-analysis of the data [70]. In the first stage of the meta-analysis, a summary statistic will be calculated for each study. In the second stage, a summary (combined) intervention effect estimate will be calculated as a weighted average of the intervention effects estimated in the individual studies. Assuming that there is heterogeneity among the studies and assuming that the intervention effects are not identical, the inverse-variance random-effects method will be considered. This involves pooling estimate of the intervention effect and its confidence interval from each included study [70]. The meta-analysis result will then be illustrated using a forest plot [70].

**4.8.3 When a meta-analysis of effect estimates is not possible.**   In case a meta-analysis of effect estimates is not possible, the effects of the intervention will be described as the size and

the direction of the observed effect and assessed through vote counting based on the direction of the effect [71]. The vote counting method based on direction of effect (rather than statistical significance) is suggested by Cochrane as an acceptable synthesis method [71]. This method enables the synthesis of diverse effect measures in systematic reviews [71].

Each effect estimates (i.e., risk ratio, odds ratio, or mean difference) will be first categorized as showing benefit or harm based on the observed direction of the effect alone, thereby creating a standardized binary metric. A count of the number of effects showing benefit will then be compared with the number showing harm. Harvest plots will provide a visual display of vote counting results.

**4.8.4 Sensitivity analysis.** Sensitivity analysis will be carried out to assess the robustness of the conclusions of this review, as well as to explore the impact of study quality on the results. This will involve a sensitivity analysis that assesses whether results change once poor-quality studies are excluded and for different designs (e.g., randomized vs non-randomized studies). Studies at higher risk of bias will be identified based on the Downs and Black's cut-off points [68] and according to the Risk-of-Bias tool (RoB 2) core features [74]: a) non-random method of allocation, b) baseline differences between intervention groups (difference between groups is big enough to result in bias), and c) outcome assessors not blinded to intervention status.

## 4.9 Ethical considerations

This study will be conducted using published data, without human involvement, therefore there is no requirement for ethical approval.

## 4.10 Dissemination

This systematic review protocol is registered in PROSPERO [CRD42023400668] to allow students, researchers, health professionals, and other collaborators to monitor its development. In addition, the findings from this review study will be published in a peer-reviewed journal.

In our complete review study, a narrative summary of the results will be provided under three subheadings: description of included (including populations and settings, interventions, comparators, and reasons for exclusion) and excluded studies (e.g., reasons for exclusion), risk of bias in included studies (including overall comments based on our sensitivity analysis and risk of bias scores), and synthesis of results (including statistical significance). In the synthesis of results section, the findings will be structured according to several key categories: population demographics, intervention characteristics, and overarching concepts. These categories will include distinctions such as results pertaining to dementia and stroke caregivers, differentiation between new and seasoned caregivers, and comparisons between telephone and video interventions, variations in intervention duration (e.g., three months versus one year).

Two tables will be created: 'characteristics of participants' (including caregiver and older adult demographics, caregiving information) and 'characteristics of included studies' (including setting and country, study design, sample size, intervention details, comparator, characteristics of outcomes, funding source, declaration of interest among primary researchers, and intervention effect estimates). Additionally, as already mentioned, a forest or harvest plot will be created depending on whether a meta-analysis is possible.

## 5. Conclusion

This systemic review will review and synthesize evidence of the effectiveness of distance interventions that aim to support preparedness of informal caregivers of older adults. Findings will inform the design of interventions that aim to support preparedness of informal caregivers of

older adults, which is essential for enhancing the health and social well-being of older people and their caregivers worldwide.

## Supporting information

**S1 Appendix. Inclusion and exclusion form.**
(DOCX)

**S2 Appendix. PRISMA-P checklist.**
(DOC)

**S3 Appendix. Search protocol.**
(DOCX)

## Author Contributions

**Conceptualization:** Fernanda L. F. Dal Pizzol, Kathleen F. Hunter, Jennifer Baumbusch, Hannah M. O'Rourke.

**Methodology:** Fernanda L. F. Dal Pizzol, Kathleen F. Hunter, Jennifer Baumbusch, Hannah M. O'Rourke.

**Project administration:** Fernanda L. F. Dal Pizzol, Kathleen F. Hunter, Hannah M. O'Rourke.

**Supervision:** Kathleen F. Hunter, Hannah M. O'Rourke.

**Validation:** Fernanda L. F. Dal Pizzol, Kathleen F. Hunter, Hannah M. O'Rourke.

**Visualization:** Fernanda L. F. Dal Pizzol, Kathleen F. Hunter, Hannah M. O'Rourke.

**Writing – original draft:** Fernanda L. F. Dal Pizzol.

**Writing – review & editing:** Fernanda L. F. Dal Pizzol, Kathleen F. Hunter, Jennifer Baumbusch, Hannah M. O'Rourke.

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
