## [Decision Letter · Decision Letter 0]

14 May 2024

PONE-D-24-11330Distance interventions for enhancing preparedness in informal caregivers of older adults: A systematic review protocolPLOS ONE

Dear Dr. Dal Pizzol,

Thank you for submitting your manuscript to PLOS ONE. After careful consideration, we feel that it has merit but does not fully meet PLOS ONE’s publication criteria as it currently stands. Therefore, we invite you to submit a revised version of the manuscript that addresses the points raised during the review process.

Both reviewers provided valuable insights for enhancing the manuscript. Reviewer 1 suggested clarifying the concept of "preparedness" compared to "readiness," refining the background to specify the focus on elderly caregivers, reconsidering inclusion criteria for study design, addressing potential age range issues in participant criteria, searching trial registries for unpublished literature, and revising search strategies to ensure comprehensive coverage. Reviewer 2 recommended considering studies where caregiver preparedness is a secondary outcome, clarifying search protocol details and incorporating "Citation Chaser" for reference tracking, expanding data extraction to include additional relevant information, describing how results will be organized and disseminated to address existing literature gaps, and justifying the necessity of the review in light of existing research, emphasizing the specificity of the topic and potential limitations of previous reviews. Incorporating these suggestions will strengthen the manuscript and provide a more comprehensive review of distance interventions for informal caregivers of older adults. Please see the detailed comments below and reply if any comment is not addressed.

We look forward to receiving your revised manuscript.

Kind regards,

Moustaq Karim Khan Rony, RN, MSS, MPH

Academic Editor

PLOS ONE

Journal Requirements:

Additional Editor Comments:

Both reviewers provided valuable insights for enhancing the manuscript. Reviewer 1 suggested clarifying the concept of "preparedness" compared to "readiness," refining the background to specify the focus on elderly caregivers, reconsidering inclusion criteria for study design, addressing potential age range issues in participant criteria, searching trial registries for unpublished literature, and revising search strategies to ensure comprehensive coverage. Reviewer 2 recommended considering studies where caregiver preparedness is a secondary outcome, clarifying search protocol details and incorporating "Citation Chaser" for reference tracking, expanding data extraction to include additional relevant information, describing how results will be organized and disseminated to address existing literature gaps, and justifying the necessity of the review in light of existing research, emphasizing the specificity of the topic and potential limitations of previous reviews. Incorporating these suggestions will strengthen the manuscript and provide a more comprehensive review of distance interventions for informal caregivers of older adults. Please see the detailed comments below and reply if any comment is not addressed.

Reviewers' comments:

Reviewer's Responses to Questions

**Comments to the Author**

1. Does the manuscript provide a valid rationale for the proposed study, with clearly identified and justified research questions?

Reviewer #1: Partly

Reviewer #2: Yes

2. Is the protocol technically sound and planned in a manner that will lead to a meaningful outcome and allow testing the stated hypotheses?

Reviewer #1: Partly

Reviewer #2: Yes

3. Is the methodology feasible and described in sufficient detail to allow the work to be replicable?

Reviewer #1: No

Reviewer #2: Yes

4. Have the authors described where all data underlying the findings will be made available when the study is complete?

Reviewer #1: Yes

Reviewer #2: Yes

5. Is the manuscript presented in an intelligible fashion and written in standard English?

Reviewer #1: Yes

Reviewer #2: Yes

6. Review Comments to the Author

You may also provide optional suggestions and comments to authors that they might find helpful in planning their study.

Reviewer #1: In “Distance interventions for enhancing preparedness in informal caregivers of older

adults: A systematic review protocol” and the authors explained the need for systematic review on this topic. I consider that the concepts described in this manuscript are worth studying but needs some revision.

#1 The concept of “preparedness” is similar with “readiness.” However, I perceived the author's intentional use of this term. I would like a little more explanation on the intent of using this term and how it differs from readiness.

#2 Background: In your background, I am not sure why you are focusing on the caregivers of “the elderly”. It looks like that your current background explains only the needs for the effect of distance interventions for informal caregiver.

#3 Inclusion criteria: You described inclusion criteria of your study as PICOS. However, I suggest you reconsider your study design framework（c, d）. You defined your inclusion criteria of study design, dealing with only RCTs or quasi-experimental design. RCTs are designed with a randomly assigned comparison group, so that they always have a comparator. Then, it is not necessary to state whether or not there is a control group. Whereas, the quasi-experimental design implies multiple study designs, of which pre-post design is one. The section [d] describes two different concepts: whether there is a comparison group and the study design. Are interrupted time series analysis and quasi-RCTs eligible? I suggest that inclusion or exclusion be defined by a more detailed description of the study design.

#4: Inclusion criteria: You defined inclusion criteria of participant as “caregivers of older adults.” If the participants in the study you will review cross the age of 60 (e.g., 50-70), how would you cope with such a case?

#5: Search protocol: You will use seven electrical databases however it does not contain trial registries. In a systematic review, I would suggest searching unpublished literature to minimize publication bias.

#6: Search strategies (Appendix3): You made search strategies as to your concern, but the concept of search strategies is too many so that your current search strategies in the row #1 might be missing out on important study. The Cochrane Handbook for systematic reviews indicates structure of a search strategy; “(i) terms to search for the health condition of interest, i.e. the population; (ii) terms to search for the intervention(s) evaluated; and (iii) terms to search for the types of study design to be included.” In the row #1 contains the way of measurement, using proximity operator. It provably rejects some noise but misses on important studies. If you want to reject some noise, I would recommend you add the row of study types in your search strategies, as Cochrane mentioned.

Reviewer #2: The research question is well-defined, and the inclusion and exclusion criteria are thorough and appropriate. Regarding the inclusion criterion "preparedness must be an intervention outcome," it raises the question that if the study did not specifically aim to intervene upon caregiver preparedness, but caregiver preparedness changed as a secondary outcome, will the study be excluded? For 4.2 search protocol, clarity is needed on the dates of the initial and any subsequent searches, ensuring that the latest search is up to date. Using “Citation Chaser” for reference and citation tracking of all included articles during data extraction is recommended. For data extraction, other information that might be meaningful to extract include year of publication, study purpose/objective, care-recipient diagnosis/severity, care-recipient age range (pediatric vs. non-pediatric), psychometrics of measurement tools/instruments that assess caregiver preparedness (if applicable), and analytical approach (if applicable). You have briefly mentioned this in the paragraph right above your aim that “Future reviews should consider how specific components of distance interventions, such as educational materials and emotional support, impact outcomes. Additionally, they need to compare different distance approaches targeting informal caregivers of older people.” To address this gap, it could be helpful if you can briefly describe how the results will be organized in 4.10 Dissemination section. For example, is there a method that helps categorize different types of intervention (caregiver-focused intervention vs. dyadic intervention; education-related intervention vs., non-education related intervention; caregiver task-focused intervention vs., caregiver emotion-focused intervention)? It will be meaningful to compare and contrast the effectiveness of interventions within each category.

While the study design, methodology, and writing style are satisfactory, there is a hesitance regarding the novelty of the topic. Session 2.1 “Why is this review needed in light of existing reviews” is not very persuasive. The three existing reviews (citation 16-18 in the manuscript) with similar topics are not very outdated (two in 2016, one in 2020). The three reviews focused on effectiveness of interventions for caregivers of patients with different diagnosis, including stroke, dementia, and heart failure. Although I agree with the author that including both formal and informal caregivers is a huge gap to address, we still need to question how helpful this review will add into the literature if we look at the effectiveness of interventions for caregivers for all kinds of diagnosis. The patient needs, caregiver routines, caregiver tasks, and emotional stress levels for different diagnosis can be very distinctive, which may lead to very different caregiver preparedness even with same intervention. Also, the previous studies “did not exclusively consider distance interventions approaches” does not seem like a strong limitation. To get the topic more focused and comparison more meaningful, we may suggest the author to consider focusing on only one type of caregivers (or caregivers with certain level of similarity besides age) but not sure if this is their goal.

7. PLOS authors have the option to publish the peer review history of their article (what does this mean?). If published, this will include your full peer review and any attached files.

Reviewer #1: No

Reviewer #2: **Yes: **Dingyue Wang

---

## [Author Response · Author response to Decision Letter 0]

24 Jun 2024

Dear Reviewers,

We would like to express our gratitude for dedicating your time and providing helpful, constructive comments. In the attached response letter, we have provided detailed responses to each of your comments. Changes made in the manuscript are tracked, and this document has also been uploaded.

Thanks,

Fernanda.

---

## [Decision Letter · Decision Letter 1]

18 Jul 2024

PONE-D-24-11330R1Distance interventions for enhancing preparedness in informal caregivers of older adults: A systematic review protocolPLOS ONE

Dear Dr. Dal Pizzol,

Thank you for submitting your manuscript to PLOS ONE. After careful consideration, we feel that it has merit but does not fully meet PLOS ONE’s publication criteria as it currently stands. Therefore, we invite you to submit a revised version of the manuscript that addresses the points raised during the review process.

Reviewer raised some more concerns about this protocol; please address these issues detailed below.

We look forward to receiving your revised manuscript.

Kind regards,

Moustaq Karim Khan Rony, RN, MSS, MPH

Academic Editor

PLOS ONE

Journal Requirements:

Reviewers' comments:

Reviewer's Responses to Questions

**Comments to the Author**

1. Does the manuscript provide a valid rationale for the proposed study, with clearly identified and justified research questions?

Reviewer #1: Partly

Reviewer #2: Yes

2. Is the protocol technically sound and planned in a manner that will lead to a meaningful outcome and allow testing the stated hypotheses?

Reviewer #1: Partly

Reviewer #2: Yes

3. Is the methodology feasible and described in sufficient detail to allow the work to be replicable?

Reviewer #1: Yes

Reviewer #2: Yes

4. Have the authors described where all data underlying the findings will be made available when the study is complete?

Reviewer #1: Yes

Reviewer #2: Yes

5. Is the manuscript presented in an intelligible fashion and written in standard English?

Reviewer #1: Yes

Reviewer #2: Yes

6. Review Comments to the Author

You may also provide optional suggestions and comments to authors that they might find helpful in planning their study.

Reviewer #1: Thank you for your careful revision. However, I could not understand only one point I have already pointed out. I would like to ask you to reconsider this point.

1. I asked you as follows;#4 Inclusion criteria: You defined inclusion criteria of participant as “caregivers of older adults.” If the participants in the study you will review cross the age of 60 (e.g., 50-70), how would you cope with such a case? And then, your reply was as follows; As stated in our revised screening form, we will not include studies that aimed to involve older adults under the age of 60. I understood this inclusion criteria. However, if your review included a study that was not age-restricted and that study conducted a subgroup analysis by age, would the results of this subgroup analysis be covered in your study? Or how would they be treated? I am sorry, but I would appreciate additional clarification.

Reviewer #2: Overall, I am pleased with the authors' response and believe this manuscript has the merit of being published. Here are a few suggestions for consideration. The authors do not need to address them now in the protocol, but the following could be helpful to think about for the full manuscript.

Database Search: It might be beneficial to search in Sociology Source Ultimate (via EBSCOhost) because caregiver preparedness is a socially constructed concept. If this search does not yield additional articles, it can be ignored. Ultimately, the authors should decide if the current seven databases are sufficient to address the research question.

Search Update: The initial search occurred in December 2023. An update before the full manuscript publication would be helpful. According to the Cochrane Handbook, "The search must be rerun close to publication if the initial search date is more than 12 months (preferably 6 months) from the intended publication date, and the results screened for potentially eligible studies."

Data Extraction: The columns for “information about caregiving” and “intervention” may contain large amounts of information, making synthesis difficult later. Consider breaking them down into more manageable pieces.

These suggestions are intended to improve the final manuscript and are not required at this stage.

7. PLOS authors have the option to publish the peer review history of their article (what does this mean?). If published, this will include your full peer review and any attached files.

Reviewer #1: No

Reviewer #2: **Yes: **Dingyue Wang

---

## [Author Response · Author response to Decision Letter 1]

31 Jul 2024

We have reviewed and addressed the reviewers' questions and suggestions. We are submitting a revised manuscript, along with an updated inclusion/exclusion form and a detailed response to the reviewers, in which we highlight the changes we made. We appreciate the reviewers' careful review and the opportunity to revise and address their concerns.

---

## [Decision Letter · Decision Letter 2]

7 Aug 2024

Distance interventions for enhancing preparedness in informal caregivers of older adults: A systematic review protocol

PONE-D-24-11330R2

Dear Dr. Dal Pizzol,

We’re pleased to inform you that your manuscript has been judged scientifically suitable for publication and will be formally accepted for publication once it meets all outstanding technical requirements.

Kind regards,

Moustaq Karim Khan Rony, RN, MSS, MPH

Academic Editor

PLOS ONE

Additional Editor Comments (optional):

Reviewers' comments:

Reviewer's Responses to Questions

**Comments to the Author**

1. Does the manuscript provide a valid rationale for the proposed study, with clearly identified and justified research questions?

Reviewer #1: Yes

2. Is the protocol technically sound and planned in a manner that will lead to a meaningful outcome and allow testing the stated hypotheses?

Reviewer #1: Yes

3. Is the methodology feasible and described in sufficient detail to allow the work to be replicable?

Reviewer #1: Yes

4. Have the authors described where all data underlying the findings will be made available when the study is complete?

Reviewer #1: Yes

5. Is the manuscript presented in an intelligible fashion and written in standard English?

Reviewer #1: Yes

6. Review Comments to the Author

You may also provide optional suggestions and comments to authors that they might find helpful in planning their study.

Reviewer #1: Protocol study must be described in a rigorous, and I believe this protocol meets that criteria.

I myself have had some trouble with the handling of age categories. I look forward to an excellent study following this protocol.

7. PLOS authors have the option to publish the peer review history of their article (what does this mean?). If published, this will include your full peer review and any attached files.

Reviewer #1: No

---

## [Editor Report · Acceptance letter]

16 Aug 2024

PONE-D-24-11330R2 

PLOS ONE

Dear Dr. Dal Pizzol, 

I'm pleased to inform you that your manuscript has been deemed suitable for publication in PLOS ONE. Congratulations! Your manuscript is now being handed over to our production team.

Kind regards, 

on behalf of

Mr. Moustaq Karim Khan Rony 

Academic Editor

PLOS ONE